# Suppression of Neuroinflammation by Coffee Component Pyrocatechol via Inhibition of NF-κB in Microglia

**DOI:** 10.3390/ijms25010316

**Published:** 2023-12-25

**Authors:** Taisuke Murata, Kenji Tago, Kota Miyata, Yasuhiro Moriwaki, Hidemi Misawa, Kenji Kobata, Yosuke Nakazawa, Hiroomi Tamura, Megumi Funakoshi-Tago

**Affiliations:** 1Division of Hygienic Chemistry, Faculty of Pharmacy, Keio University, 1-5-30 Shibakoen, Minato-ku 105-8512, Tokyo, Japan; penpenmosa@keio.jp (T.M.); nakazawa-ys@pha.keio.ac.jp (Y.N.); tamura-hr@keio.jp (H.T.); 2Department of Laboratory Sciences, Gunma University Graduate School of Health Sciences, 3-39-22 Showa-Machi, Maebashi 371-8514, Gunma, Japan; ktago@gunma-u.ac.jp; 3Division of Pharmacology, Faculty of Pharmacy, Keio University, 1-5-30 Shibakoen, Minato-ku 105-8512, Tokyo, Japan; miyata_006499@keio.jp (K.M.); moriwaki-ys@pha.keio.ac.jp (Y.M.); misawa-hd@pha.keio.ac.jp (H.M.); 4Department of Pharmaceutical Science, Josai University, 1-1 Keyakidai, Sakado 350-0295, Saitama, Japan; kobata@josai.ac.jp

**Keywords:** coffee, pyrocatechol, neuroinflammation, microglia, NF-κB

## Abstract

According to numerous studies, it has been epidemiologically suggested that habitual coffee intake seems to prevent the onset of neurodegenerative diseases. In this study, we hypothesized that coffee consumption suppresses neuroinflammation, which is closely related to the development of neurodegenerative diseases. Using microglial BV-2 cells, we first found that the inflammatory responses induced by lipopolysaccharide (LPS) stimulation was diminished by both coffee and decaffeinated coffee through the inhibition of an inflammation-related transcription factor, nuclear factor-κB (NF-κB). Pyrocatechol, a component of roasted coffee produced by the thermal decomposition of chlorogenic acid, also exhibited anti-inflammatory activity by inhibiting the LPS-induced activation of NF-κB. Finally, in an inflammation model using mice injected with LPS into the cerebrum, we observed that intake of pyrocatechol as well as coffee decoctions drastically suppressed the accumulation of microglia and the expression of interleukin-6 (IL-6), tumor necrosis factor α (TNFα), CCL2, and CXCL1 in the inflammatory brain. These observations strongly encourage us to hypothesize that the anti-inflammatory activity of pyrocatechol as well as coffee decoction would be useful for the suppression of neurodegeneration and the prevention of the onsets of Alzheimer’s (AD) and Perkinson’s diseases (PD).

## 1. Introduction

Microglia are immune cells that are resident in the brain and contribute to the maintenance of the microenvironment of the central nervous system (CNS) [1]. Microglia originate from yolk sac macrophages and migrate to the brain before the formation of the blood brain barrier [2]. Although microglia are similar cells to macrophages in several characteristics such as cell surface markers and expressed cytokines, the role of microglia is very different from that of macrophages. Under physiological conditions, microglia are involved in neuronal development and also maintain homeostasis in the CNS through surveillance of the microenvironment of the brain with a ramified morphology [3]. Neuronal injury or an inflammation-mimicked experimental stimulation such as lipopolysaccharide (LPS) activates microglial cells with an amoeboid morphology [4]. Activated microglial cells secrete proinflammatory cytokines and chemokines including interleukin-6 (IL-6), tumor necrosis factor α (TNFα), CCL2 and CXCL1. Via the secretion of these cytokines and chemokines, other immune cells are recruited to inflammatory regions and neuronal cell injury sites [5,6]. Activated microglia also induce the expression of inducible nitric oxide synthase (iNOS), an enzyme catalyzing the production of nitric oxide (NO) exhibiting neurotoxicity [7,8,9]. Furthermore, microglia produce a principal inflammatory mediator, prostaglandin E2 (PGE2), through the expression of cyclooxygenase-2 (COX-2) [10].

The production of inflammation-related mediators in microglia is induced by the Toll-like receptor 4 (TLR4)-mediated activation of transcription factors, such as nuclear factor-κB (NF-κB) and activator protein 1 (AP-1) [11,12,13,14]. The NF-κB subunits p50 and p65 form a heterodimer and reside in the cytoplasm by binding to inhibitor kappa B alpha (IκBα) [15]. The phosphorylation of IκBα by activated IκB kinase and its subsequent degradation induces the nuclear translocation and transcriptional activation of NF-κB [16]. On the other hand, AP-1 is a dimeric transcription factor formed from Jun and Fos families, and AP-1 is activated by mitogen-activated protein kinase (MAPK) family members, including extracellular signal-regulated kinase (ERK) and c-Jun N terminal kinase (JNK) [17]. These immune responses are tightly controlled in microglia to maintain tissue homeostasis.

In contrast, chronic neuroinflammation has been associated with the pathology of several neurodegenerative diseases, including Alzheimer’s disease (AD) and Parkinson’s disease (PD). Infiltration of activated microglia was observed in the substantia nigra of the postmortem brains of patients with AD and PD [18,19,20]. Furthermore, the expression and production of pro-inflammatory cytokines, chemokines, and NO were significantly induced in the cerebrospinal fluid and serum of patients with AD and PD. These events indicate that neuroinflammation is an early characteristic for the onsets of neurodegenerative diseases including AD and PD [21,22,23]. Recently, AD and PD model mice have been utilized for several studies. Among them, mice harboring APPswe/PS1M146V/tauP301L are utilized as AD model mice. When these AD model mice were treated with a TNFα inhibitor, etanercept, the level of neuron injury and production of TNFα, IL-6, and CCL2 decreased and spatial, long-term, and working memory improved [24]. Sriram et al. also reported that TNFα signaling pathway is essential for 1-methyl-4-phenyl pyridinium (MPP)-induced dopaminergic neurotoxicity [25]. Furthermore, the non-steroidal anti-inflammatory drug, Ibuprofen, was previously shown to be effective in the treatment of neuroinflammation associated with β-amyloid (Aβ) deposition and PD-like symptoms [26,27]. Therefore, the inhibition of neuroinflammation is expected to be effective as a therapeutic strategy for neurodegenerative diseases.

Coffee is one of the most popular beverages worldwide and contains a range of bioactive compounds, including caffeine, chlorogenic acid, and polyphenols [28,29]. Epidemiological studies previously demonstrated the protective role of coffee, which decreased the risk of developing AD and PD [30,31]. It was reported that habitual intake of coffee in middle age is associated with a lower risk of developing subsequent dementia and AD than for those who drank no or only a small amount of coffee. The lowest risk of developing dementia and AD was among those who drank three to five cups of coffee per day, with a 65% reduction in risk [30]. Furthermore, it was also reported that the increase in coffee intake consistently decreases the incidence of PD [31]. However, it remains to be clarified how coffee intake suppresses the onset of AD and PD and which components in coffee contribute to the preventive effect of coffee.

We previously found the presence of anti-inflammatory effects in coffee decoction against murine macrophage cell line RAW264.7 [32]; therefore, we hypothesized that coffee may also be effective for the suppression of neuroinflammation. Therefore, we first tested whether the anti-inflammatory activities of coffee and decaffeinated coffee decoctions could be effective on LPS-induced neuroinflammation in murine microglial BV-2 cells. We also investigated the effects of coffee and decaffeinated coffee intakes on neural inflammation in mice triggered by an injection with LPS. The results obtained demonstrated that coffee intake attenuated neuroinflammation, and its anti-neuroinflammatory activity was attributed to the intake of pyrocatechol.

## 2. Results

### 2.1. Coffee and Decaffeinated Coffee Both Inhibited LPS-Induced Inflammatory Responses in BV-2 Cells

To assess the anti-inflammatory activities of coffee and decaffeinated coffee decoctions, we investigated their effects on LPS-induced neuroinflammation in the murine microglia cell line BV-2. We initially confirmed that the viability of BV-2 cells was not affected by the coffee or decaffeinated coffee decoction of up to 5% (*v*/*v*) regardless of the LPS stimulation (Figure 1A). Both coffee and decaffeinated coffee decoctions dose-dependently inhibited LPS-induced NO production. (Figure 1B). We then investigated the effects of the decoctions on the expression of iNOS, which induces the production of NO [9] by RT-PCR and immunoblotting. The LPS-induced mRNA and protein expression of iNOS was significantly inhibited by the coffee and decaffeinated coffee decoctions (Figure 1C,D). These results suggest that the coffee decoction significantly inhibited LPS-induced NO production by preventing iNOS mRNA expression in BV-2 cells and that ingredients other than caffeine may have attenuated neuroinflammation.

We also examined the effects of coffee and decaffeinated coffee decoctions on the LPS-induced expression of cytokines, chemokines, and COX-2 in BV-2 cells. The expression of IL-6 mRNA was induced from 2 h after the LPS stimulation, peaked at 6 h, and decreased at 8 h, whereas the expression of TNFα mRNA peaked 1 h after the LPS stimulation and then decreased. In addition, the expression of CCL2 mRNA increased up to 6 h after the LPS stimulation and then decreased, while the expression of CXCL1 mRNA and COX-2 mRNA peaked 2 h after the LPS stimulation and then decreased (Figure 2A–E). The coffee and decaffeinated coffee decoctions both markedly attenuated the LPS-induced mRNA expression of IL-6, TNFα, CCL2, CXCL1, and COX-2 (Figure 2A–D). These results revealed that the coffee and decaffeinated coffee decoctions strongly suppressed neuroinflammation.

### 2.2. Coffee and Decaffeinated Coffee Both Inhibited the LPS-Induced Nuclear Translocation and Transcriptional Activation of NF-κB in BV-2 Cells

We examined the effects of the coffee and decaffeinated coffee decoctions on the LPS-induced transcriptional activation of NF-κB using a luciferase assay. Both decoctions significantly inhibited the LPS-induced transcriptional activation of NF-κB in BV-2 cells (Figure 3A). The LPS-induced degradation of IκBα is a trigger for the nuclear translocation of NF-κB, and these steps are required for the transcriptional activation of NF-κB [15,16]. However, neither the coffee nor decaffeinated coffee decoction affected the LPS-induced degradation of IκBα (Figure 3B). To clarify the effects of the coffee and decaffeinated coffee decoctions on the LPS-induced nuclear translocation of NF-κB, we performed immunofluorescence staining for the NF-κB p65 subunit, which is a major subunit of LPS-activated NF-κB. As shown in Figure 3C, the coffee decoction effectively inhibited the LPS-induced nuclear translocation of NF-κB p65. Therefore, the coffee and decaffeinated coffee decoctions appeared to inhibit the LPS-induced nuclear localization and subsequent activation of NF-κB without affecting the degradation of IκBα. 

LPS also induces the activation of MAPK family members, such as ERK, p38, and JNK, which are involved in inflammatory responses [12,13,33,34]. Therefore, the effects of the coffee and decaffeinated coffee decoctions on the LPS-induced phosphorylation of ERK, p38, and JNK were assessed by immunoblotting. However, neither the coffee nor decaffeinated coffee decoction inhibited the LPS-induced phosphorylation of ERK, p38, or JNK in BV-2 cells (Figure 4). Collectively, these results suggest that coffee and decaffeinated coffee exhibited anti-inflammatory activity by specifically inhibiting the NF-κB pathway in LPS signaling.

### 2.3. The Coffee Ingredient, Pyrocatechol Inhibited LPS-Induced Inflammatory Responses in BV-2 Cells

Since decaffeinated coffee suppressed neuroinflammation in BV-2 cells to a similar extent as coffee (Figure 1 and Figure 2), components in the coffee decoction other than caffeine exhibit anti-inflammatory activity. Chlorogenic acid, one of the major components of coffee beans, is thermally decomposed by roasting and is present in negligible amounts in roasted coffee. The thermal decomposition of chlorogenic acid results in the production of pyrocatechol (Figure 5A) [29,35]. HPLC analysis detected pyrocatechol, not only in coffee but also in decaffeinated coffee at approximately 0.1 mM [29]. Since we previously reported that pyrocatechol exhibited anti-inflammatory activity in LPS-stimulated macrophages [32], we speculated that pyrocatechol may also exert anti-neuroinflammatory effects in microglial cells. Therefore, we investigated the effects of pyrocatechol on LPS-induced neuroinflammation using pyrocatechol at similar concentrations to those present in the coffee and decaffeinated coffee decoctions used in the above experiments. The results obtained confirmed that pyrocatechol up to 10 μM did not affect the viability of BV-2 cells regardless of the LPS stimulation (Figure 5B). Pyrocatechol inhibited LPS-induced NO production in a dose-dependent manner (Figure 5C). It also significantly inhibited the LPS-induced mRNA and protein expression of iNOS in BV-2 cells (Figure 5D,E). Furthermore, the results of examining the concentration of pyrocatechol showed that above 0.6 μM, pyrocatechol significantly inhibited LPS-induced NO production, and that pyrocatechol up to 100 μM did not exhibit cytotoxicity after 12 h of treatment. 

We then examined the effects of pyrocatechol on the LPS-induced expression of cytokines, chemokines, and COX-2 in BV-2 cells. Pyrocatechol significantly inhibited the LPS-induced mRNA expression of TNFα, IL-6, CCL2, CXCL1, and COX-2 (Figure 6A–E). These results demonstrated that pyrocatechol is one of the ingredients in coffee that contributes to its inhibitory effects on neuroinflammation.

### 2.4. The Coffee Ingredient, Pyrocatechol, Inhibited the LPS-Induced Nuclear Translocation and Transcriptional Activation of NF-κB in BV-2 Cells

We also investigated the effects of pyrocatechol on LPS-induced NF-κB activation. Pyrocatechol significantly inhibited the LPS-induced transcriptional activation of NF-κB in BV-2 cells (Figure 7A). Similar to the coffee and decaffeinated coffee decoctions, pyrocatechol did not affect the degradation of IκBα by LPS but inhibited the nuclear translocation of the NF-κB p65 subunit (Figure 7B,C).

Furthermore, pyrocatechol failed to suppress the LPS-induced activation of ERK, JNK, and p38, similar to the coffee and decaffeinated coffee decoctions (Figure 8). These results suggest that pyrocatechol is a coffee component that exhibits anti-inflammatory activity through the specific inhibition of NF-κB activation in BV-2 cells.

### 2.5. Coffee and Decaffeinated Coffee Both Attenuated the LPS-Induced Accumulation of Microglia and mRNA Expression of IL-6, TNFα, CCL2, and CXCL1 in Mice

The effects of the intake of coffee and decaffeinated coffee on neuroinflammation were assessed using C57/BL6 mice. We previously reported that C57/BL6 mice drank up to 60% (*v*/*v*) coffee to a similar extent as regular water [36]. Therefore, C57BL/6 mice were given water as the control, a 60% (*v*/*v*) coffee decoction, or 60% (*v*/*v*) decaffeinated coffee decoction for 4 weeks. No significant changes were observed in the body weight or food intake of mice with the intake of water, coffee, or decaffeinated coffee (Figure 9A). In mice that drank coffee or decaffeinated coffee, LPS was injected into the right striatum to induce local neuroinflammation. Iba1 is constitutively expressed in microglia and is widely used as a classical protein marker specific to microglia in the brain [37]. Murine brain sections were prepared and immunostained with an anti-Iba1 antibody to detect microglia. The LPS injection into the right striatum induced the accumulation of microglia in the murine right brain. The intake of coffee and decaffeinated coffee both significantly suppressed the LPS-induced accumulation of microglia in the right brain (Figure 9B). 

We also investigated the effects of the intake of coffee and decaffeinated coffee on the LPS-induced expression of inflammatory mediators in the murine brain by RT-PCR. The LPS injection strongly induced the mRNA expression of IL-6, TNFα, CCL2, and CXCL1 in the murine brain, which was significantly attenuated by the intake of coffee and decaffeinated coffee (Figure 9C). These results suggest that the intake of coffee and decaffeinated coffee attenuated neuroinflammation by preventing the expression of inflammatory mediators in vivo.

### 2.6. The Intake of Pyrocatechol Attenuated the LPS-Induced Accumulation of Microglia and mRNA Expression of IL-6, TNFα, CCL2, and CXCL1 in Mice

We investigated whether the anti-neuroinflammatory effects of coffee consumption were due to the intake of pyrocatechol. The concentration of pyrocatechol in 60% (*v*/*v*) coffee and decaffeinated coffee was 58.3–74 µM [29,32]. C57BL/6 mice were given water as the control or 60 μM pyrocatechol solution for 4 weeks. No significant changes were observed in the body weight or food intake of mice due to the intake of pyrocatechol (Figure 10A). After the administration of pyrocatechol, LPS was injected into the right striatum to induce neuroinflammation. The immunostaining analysis of the murine brain with anti-Iba1 antibody revealed that the intake of pyrocatechol significantly suppressed the LPS-induced accumulation of microglia in the brain (Figure 10B). We also examined the effects of the intake of pyrocatechol on the LPS-induced expression of inflammatory mediators in the murine brain using RT-PCR. The intake of pyrocatechol significantly attenuated the LPS-induced mRNA expression of IL-6, TNFα, CCL2, and CXCL1 in the murine brain (Figure 10C). Collectively, our present study demonstrated that pyrocatechol is responsible for the inhibitory effects of coffee consumption on LPS-induced neuroinflammation, and this is mediated by the suppression of NF-κB (Figure 11). 

## 3. Discussion

To date, many epidemiological studies have reported the effects of coffee drinking on human health. It has been reported that coffee consumption is well correlated with the prevention of onsets of various types of cancers including colorectal cancer and pancreatic cancer [38,39]. It was also shown that coffee consumption enhances the efficiency of the treatment with tamoxifen to breast cancer patients [40]. We found that coffee decoction accelerated the ability of tamoxifen to induce the apoptotic cell death in breast cancer cell line MCF-7 cells through the arrest of cell cycle and activation of p53 tumor suppressor [41]. In addition, we also reported that coffee decoction possesses the ability to suppress insulin-caused adipocyte differentiation [36], suggesting that coffee may also be effective in the prevention of obesity.

Epidemiological studies also have suggested that habitual consumption of coffee seems to prevent the onset and progression of neurodegenerative diseases including AD and PD [30,31]; however, its molecular mechanisms remain unknown. To induce neuroinflammation in microglial BV-2 cells and mice, we utilized LPS as an experimental stimulator of TLR4. The pathogenesis of neurodegeneration in AD and PD is driven by the abnormal accumulation of denatured Aβ and α-synuclein in the CNS, respectively [42,43]. Aβ aggregates and α-synuclein oligomers both trigger the production of a number of proinflammatory cytokines from microglia through TLR4, which is expressed on the cell surface of microglia [44,45]. For these reasons, the LPS-induced neuroinflammation model used in the present study is expected to reflect neuroinflammation in AD and PD. We showed that the treatments with the coffee and decaffeinated coffee decoctions markedly inhibited LPS-induced inflammatory responses in BV-2 cells (Figure 1 and Figure 2) and that coffee intake significantly attenuated neuroinflammation in mice intracerebrally injected with LPS (Figure 9). Therefore, the inhibitory effects of coffee on neuroinflammation may contribute to preventing the onset and progression of neurodegenerative diseases.

To elucidate the molecular mechanisms underlying the anti-inflammatory effects of coffee, we also attempted to clarify the effects of coffee on the LPS signaling pathway and identify the coffee components exhibiting anti-inflammatory activity. Although LPS activates the NF-κB and MAPK pathways via TLR4, the coffee decoctions specifically inhibited the NF-κB pathway in BV-2 cells (Figure 3 and Figure 4). We showed that coffee decoction suppressed the expression of LPS-induced expression of NF-κB-target genes such as IL-6, TNFα, CCL2, CXCL1, and COX-2 in BV-2 cells (Figure 2 and Figure 6). On the other hand, we observed that the expression of interleukin-10 (IL-10), which has the function of suppressing inflammatory responses, was hardly changed by LPS stimulation or the treatment with coffee decoction or pyrocatechol [32], suggesting that the anti-inflammatory activity of coffee decoction is due to inhibition of NF-κB activity. The coffee decoction did not inhibit LPS-induced IκBα degradation but significantly attenuated the LPS-induced nuclear translocation of NF-κB and its transcriptional activity (Figure 3). In unstimulated cells, the NF-κB heterodimer p65/p50 is retained by IκBα in the cytoplasm and masks the p65-nuclear localization sequence (NLS) [13,14]. Once IκBα is degraded by the LPS stimulation, p65-NLS is exposed and the NF-κB heterodimer p65/p50 is translocated into the nucleus by adapter importin α3 and receptor importin β [46]. Therefore, coffee may specifically suppress the nuclear translocation of NF-κB by inhibiting the function of importins α3 and β or binding between NF-κB and importins. 

The present results also revealed that pyrocatechol is one of the ingredients in coffee that exhibits anti-inflammatory activity through the same mechanism as the coffee decoction (Figure 5, Figure 6, Figure 7 and Figure 10). We previously reported that anti-inflammatory activity was detected in the murine macrophage cell line, RAW264.7, treated with a decoction of roasted coffee beans but was absent in cells treated with a decoction of green coffee beans, indicating that the components produced in coffee beans by roasting exhibit anti-inflammatory activity [32]. Chlorogenic acid, which is present in green coffee beans, is unstable to heat and degraded by the roasting procedure, resulting in the formation of caffeic acid, quinic acid, pyrocatechol, and 4-ethylcatechol [29,35]. Therefore, not only pyrocatechol, but also these chlorogenic acid degradation products are expected to exhibit anti-inflammatory activity. In the future, it will be possible to fully elucidate the molecular mechanisms underlying the inhibitory effects of coffee on neuroinflammation by examining the contents and effects of other chlorogenic acid degradation products. Previous studies reported that pyrocatechol exhibited acute toxicity and mutagenicity, which could cause oncogenicity [47]. Although we understand that the concentration of pyrocatechol utilized in our study did not exhibit toxicity to the mice, we still need to confirm its mutagenicity. To overcome this problem, we need to chemically modify pyrocatechol to develop safe derivatives. Furthermore, we have not yet investigated the potential off-target effects of pyrocatechol on cellular processes beyond inflammation. In this study, we clarified that (1) coffee and pyrocatechol did not affect the cell growth and viability of BV-2 cells as shown in Figure 1 and Figure 5 and that (2) coffee and pyrocatechol suppressed the nuclear translocation and transcriptional activation of NF-κB. In the previous study, we additionally clarified that the inhibitory effect of pyrocatechol on the activation of NF-κB was due to the pyrocatechol-induced activation of the Keap1-Nrf2 pathway in macrophages [32]. It is quite simple speculation that similar machinery regulating NF-κB activation may also work in BV-2 cells. Several studies have reported that oxidative-stress-induced activation of Nrf2 most likely contributes to the prevention of not only neural inflammation but also tumorigenesis and tumor progression [48,49]. Inflammation-related cells such as macrophages are well known to present in tumor microenvironments and contribute to tumor progression [50]. Accordingly, although it may be too speculative, coffee decoction and its ingredients such as pyrocatechol may also suppress tumorigenesis or tumor progression.

Recently, several studies also reported that ingredients or their derivatives, derived from foods or plants, suppress neural inflammation responses in microglia or their related neurodegenerative diseases. Using in vitro and in vivo experimental models, Bao and colleagues reported that shikimic acid suppressed LPS-induced neuroinflammation through the inhibition of NF-κB activation [51]. They further described that shikimic acid activated Nrf2 pathway to inhibit NF-κB activation. Li et al. also clarified that neferine exhibited anti-inflammatory activity in BV-2 cells through inhibition of LPS-induced phosphorylation and nuclear translocation of the NF-κB p65 subunit [52]. Compared to these reports, we also clarified that pyrocatechol suppresses the nuclear translocation and the transcriptional activation of NF-κB in LPS-stimulated BV-2 cells. For the activation of Nrf2, inactivation of ubiquitination machinery including KEAP1 is known to be essential, and we also clarified that inactivation of KEAP1 is an indispensable step for pyrocatechol-induced activation of Nrf2 [53]. In our study, it is still unclear how pyrocatechol suppresses the function of NLS of the NF-κB p65 subunit and how pyrocatechol causes the inactivation of KEAP1. These should be clarified to understand the biological activities in coffee decoctions in future investigations.

In the present study, we investigated the LPS-treated microglial cell line BV-2 and mice injected with LPS into the brain to show that coffee and its component, pyrocatechol, suppressed neuroinflammation (Figure 11). However, it is considered that actual neural inflammations related to neurodegenerative diseases including AD and PD are triggered by the accumulation of aggregated Aβ or α-synuclein [42,43]. In our current experimental system, it is difficult to test whether pyrocatechol is also effective on neural inflammation caused by Aβ or α-synuclein, and this issue needs to be resolved in the future. 

## 4. Materials and Methods

### 4.1. Reagents and Antibodies

Lipopolysaccaride (LPS) derived from *Escherichia coli* (O111:B4) was obtained from InvivoGen (San Diego, CA, USA). For preparation of coffee and decaffeinated coffee decoctions, 8 g of coffee beans ground as powder (Columbia Arabica, Starbucks Coffee, Tokyo, Japan) was mixed with 140 mL of 95 °C hot water, and the decoction was obtained through a paper filter [34]. These prepared coffee decoctions were defined as 100% (*v/v*). Anti-iNOS, anti-β-actin, anti-IκBα, anti-NF-κB p65, and anti-p38 antibodies were obtained from Santa Cruz Biotechnology Inc. (Santa Cruz, CA, USA). An anti-Iba1 antibody was purchased from FUJIFILM Wako Pure Chemical Corporation (Osaka, Japan). Other antibodies were purchased from Cell Signaling Technology (Danvers, MA, USA). Other chemicals were purchased from Nacalai Tesque (Kyoto, Japan).

### 4.2. Cell Culture

BV-2 mouse microglial cells were purchased from Banca Biologica e Cell Factory (Genova, Italy) and cultured in Dulbecco’s Modified Eagle’s Medium (DMEM) (Nacalai Tesque) supplemented with 10% heat-inactivated fetal bovine serum (Biowest, Nuaillé, France), 100 units/mL penicillin (Nacalai Tesque), and 100 μg/mL streptomycin (Nacalai Tesque) at 37 °C and 5% CO_2_ in a humidified incubator. 

### 4.3. Measurement of Cell Viability and NO Production

BV-2 cells (2 × 10^5^ cells) were seeded on 24-well plates and pretreated with coffee or decaffeinated coffee (0.3125, 0.625, 1.25, 2.5, 5, and 10% (*v/v*)) or pyrocatechol (2.5, 5, and 10 μM) for 1 h followed by stimulation with LPS (1 μg/mL) for 12 h. Cell viability was assessed using the trypan blue dye exclusion test. The culture medium was collected, and nitrite was measured by the Griess reaction as previously described [32]. 

### 4.4. SDS-Polyacrylamide Gel Electrophoresis (PAGE) and Immunoblotting

Immunoblotting was performed as previously described [32]. Polyvinylidene difluoride membranes (Millipore, Billerica, MA, USA) were incubated with primary and horseradish peroxidase-conjugated secondary antibodies and visualized with the ECL detection system (Cytiva, Marlborough, MA, USA). Band intensities were measured using ImageJ software (https://imagej.net/software/imagej). The relative expression levels and phosphorylation levels of each molecule were shown in the graphs. Each image in each figure is representative of three independent experimental results.

### 4.5. Reverse Transcription–Polymerase Chain Reaction (RT-PCR)

To prepare total RNA, Sepasol (Nacalai Tesque) was added to BV-2 cells and the right striatum of each mouse was injected with LPS, and RNA fractions were extracted by following protocol. Complementary DNA (cDNA) was prepared by reverse transcriptional reaction using ReverTraAce and an oligo (dT)_20_ primer (Toyobo Life Science, Inc., Tokyo, Japan). The expression of interested gene expression was analyzed by quantitative real-time PCR using an ABI 7300 thermal cycler and Luna Universal qPCR Master Mix (New England Biolabs, Ipswich, MA, USA) as previously described [32]. The PCR primer sequences were as follows: iNOS 5′-TCTGCGCCTTTGCTCATGAC-3′(Upstream) and 5′-TAAAGGCTCCGGGCTC-3′ (downstream); IL-6 5′-CCACTTCACAAGTCGGAGGC-3′ (Upstream) and 5′-GGAGAGCATTGGAAATTGGGGT-3′ (downstream); TNFα 5′-TACTGAACTTCGGGGTGATCGG-3′ (Upstream) and 5′-CAGCCTTGTCCCTTGAAGAGAA-3′ (downstream); CCL2 5′-TGAGGTGGTTGTGGAAAAGG-3′ (Upstream) and 5′-CCTGCTGTTCACAGTTGCC-3′ (downstream); CXCL1 5′-GCCTATCGCCAATGAGCTG-3′ (Upstream) and 5′-TGGGGACACCTTTTAGCATC-3′ (downstream); *COX-2* 5′-*AGAAGGAAATGGCTGCAGAA*-3′ (Upstream) and 5′-GCTCGGCTTCCAGTATTGAG-3′ (downstream); GAPDH 5′-ACTCCACTCACGGCAAATTC-3′ (Upstream) and 5′-CCTTCCACAATGCCAAAGTT-3′ (downstream). The expression of GAPDH mRNA was utilized as an internal control.

### 4.6. Transfection and Luciferase Assays

BV-2 cells (2 × 10^6^ cells) were seeded on 6-well plates and transfected with 2 μg of pNF-κB-Luc and 2 μg of pRL-TK using 10 μL of 1 μg/mL polyethyleneimine Max (Polysciences, Inc., Warrington, PA, USA). Twelve hours after transfection, the medium was changed to fresh culture medium and pretreated with coffee (5% (*v/v*)) and decaffeinated coffee (5% (*v/v*)) decoctions or pyrocatechol (5 μM) for 1 h followed by a stimulation of LPS (1 μg/mL) for 8 h. The luciferase assay was performed using the Dual-Luciferase Reporter Assay System (Promega, Madison, WI, USA).

### 4.7. Immunofluorescence Staining

BV-2 cells were plated on sterile coverslips and pretreated with coffee (5% (*v/v*)), decaffeinated coffee (5% (*v/v*)), or pyrocatechol (5 μM) for 1 h followed by a stimulation with LPS for 30 min. After cells were fixed in 4% paraformaldehyde in PBS (pH7.4) and permeabilized in 0.2% (*v/v*) Triton X-100 in PBS, the cells were immunostained using anti-NF-κB p65 antibody and Alexa Fluor^®^ 488 goat anti-mouse IgG (Invitrogen, Carlsbad, CA, USA) as previously reported [54]. The sections of murine brain were prepared and immunofluorescence staining using anti-Iba1 antibody and Alexa FluorTM488 goat anti-rabbit IgG (Invitrogen) was performed as previously reported [54]. Fluorescent images were captured by an all-in-one microscope (Keyence, Osaka, Japan), and 20 images were merged. 

### 4.8. Induction of Neuroinflammation in Mice Administered LPS

Male C57BL/6JJmsSlc mice (4 weeks old) were obtained from Sankyo Labo Service Corporation, Inc. (Tokyo, Japan). Mice were given water, a 60% (*v/v*) coffee decoction, 60% (*v/v*) decaffeinated coffee decoction, or 60 μM pyrocatechol for 4 weeks. Mice were allowed free access to food, water, and the coffee decoctions. Body weight and food intake were measured twice a week for 4 weeks. Intracerebral injection of LPS into mice was performed as previously described [54]. Briefly, after anesthetized mice were fixed to the stereotactic fixation device, SR-6M-HT (NARISHIGE, Tokyo Japan) at anteroposterior + 0.2 mm, mediolateral + 2.0 mm from the bregma, a hole was drilled into the skull. LPS was injected at a rate of 0.6 μL per 3 min using a 5-μL Hamilton syringe. All experimental protocols were approved by the Animal Usage Committee of Keio University (Approval number, A2022-294). All animal experiments were performed in accordance with relevant guidelines and regulations (Institutional Guidelines on Animal Experimentation at Keio University). This study is reported in accordance with ARRIVE guidelines.

### 4.9. Statistical Analysis

Each experiment was performed at least three times, and all data were expressed as the mean ± standard deviation (SD). A one-way analysis of variance (ANOVA) was used to evaluate the significance of differences using Prism 8.0.1 (GraphPad, San Diego, CA, USA), and a *p*-value < 0.05 was considered to be significant.

## 5. Conclusions

Pyrocatechol is an ingredient in both coffee and decaffeinated coffee. It significantly suppresses the LPS-induced production of inflammation-related mediators, such as NO, cytokines, and chemokines, by inhibiting NF-κB activation in microglia. Therefore, the intake of coffee or decaffeinated coffee suppresses neuroinflammation and is expected to effectively prevent the onset of various neurodegenerative diseases.

## Figures and Tables

**Figure 1 ijms-25-00316-f001:**
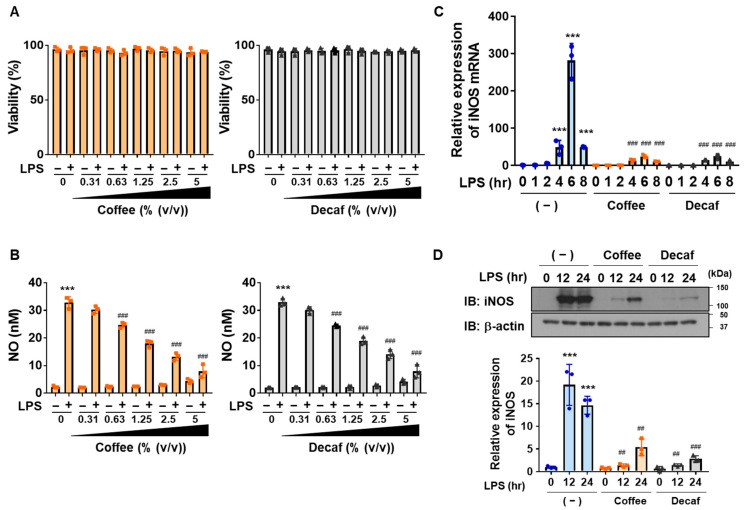
Coffee and decaffeinated coffee both inhibit LPS-induced NO production and expression of iNOS in BV-2 cells. (**A**,**B**) BV-2 cells were pretreated with coffee or decaffeinated coffee (Decaf) (0.31, 0.63, 1.25, 2.5, and 5% (*v*/*v*)) for 1 h followed by stimulation with LPS (1 μg/mL) for 12 h. (**A**) Cell viability was measured. (**B**) The amount of NO in the culture supernatant was measured. (**C**,**D**) BV-2 cells were pretreated with coffee or decaffeinated coffee (Decaf) (5% (*v*/*v*)) for 1 h followed by stimulation with LPS (1 μg/mL) for the indicated periods. (**C**) The expression of iNOS mRNA was analyzed by RT-PCR. (**D**) The expression of iNOS was examined by immunoblotting. The relative expression level of iNOS is shown in the graph. (**A**–**D**) Values are the mean ± S.D. of three independent experiments. *** *p* < 0.001 indicates significant difference from control cells. ^##^ *p* < 0.01 and ^###^ *p* < 0.001 indicate significant differences from control cells treated with LPS.

**Figure 2 ijms-25-00316-f002:**
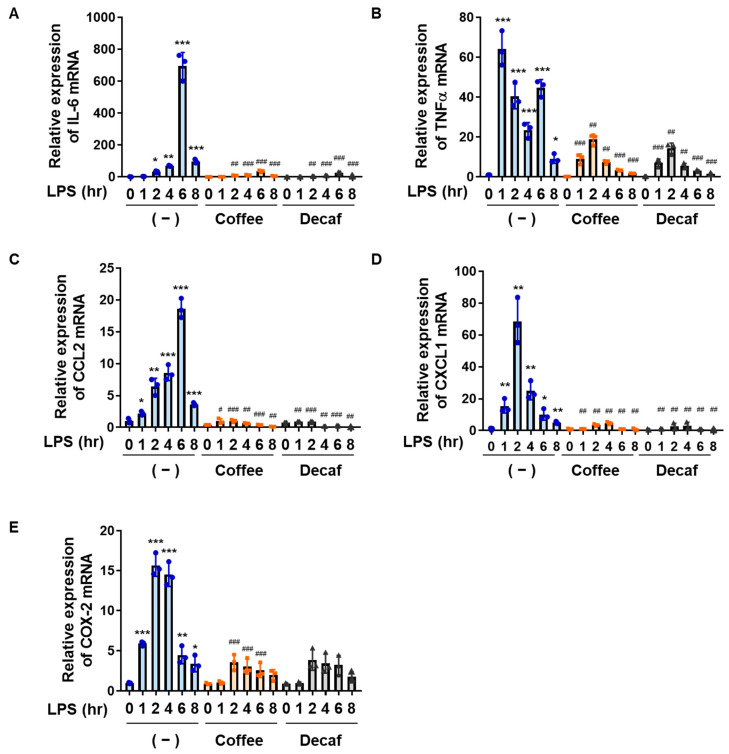
Coffee and decaffeinated coffee both inhibit LPS-induced expression of inflammatory mediators in BV-2 cells. BV-2 cells were pretreated with coffee or decaffeinated coffee (Decaf) (5% (*v*/*v*)) for 1 h followed by stimulation with LPS (1 μg/mL) for the indicated periods. The mRNA expression of (**A**) IL-6, (**B**) TNFα, (**C**) CCL2, (**D**) CXCL1, and (**E**) COX-2 was analyzed by RT-PCR. Values are the mean ± S.D. of three independent experiments. * *p* < 0.05, ** *p* < 0.01, and *** *p* < 0.001 indicate significant differences from control cells. ^#^
*p* < 0.05, ^##^
*p* < 0.01, and ^###^
*p* < 0.001 indicate significant differences from control cells treated with LPS.

**Figure 3 ijms-25-00316-f003:**
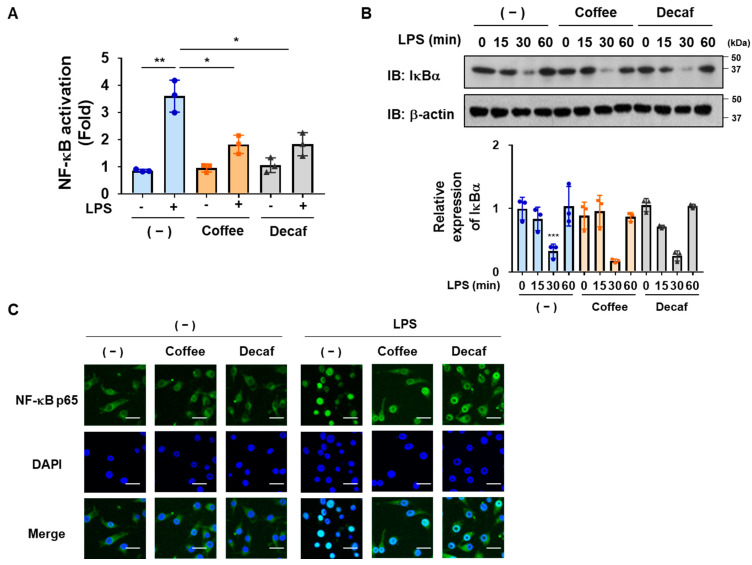
Coffee and decaffeinated coffee both inhibit the LPS-induced nuclear localization and transcriptional activation of NF-κB in BV-2 cells. (**A**) BV-2 cells were transfected with pNF-κB-Luc and pRL-TK as described in the Methods. Cells were pretreated with coffee or decaffeinated coffee (Decaf) decoctions (5% (*v*/*v*)) for 1 h followed by stimulation with LPS (1 μg/mL) for 8 h. Luciferase assay was performed. Values are the mean ± S.D. of three independent experiments. * and ** indicate *p* < 0.05 and *p* < 0.01, respectively. (**B**) BV2 cells were pretreated with coffee or decaffeinated coffee (Decaf) decoctions (5% (*v*/*v*)) for 1 h followed by stimulation with LPS (1 μg/mL) for indicated periods. The expression of IκBα was examined by immunoblotting. The relative expression level of IκBα is shown in the graphs. Values are the mean ± S.D. of three independent experiments. *** *p* < 0.001 indicates a significant difference from control cells. (**C**) BV-2 cells were pretreated with coffee or decaffeinated coffee (Decaf) decoctions (5% (*v*/*v*)) for 1 h followed by stimulation with LPS (1 μg/mL) for 30 min. The localization of NF-κB p65 was visualized with an antibody (green). DAPI (blue) was also applied to visualize nuclei. (Scale bar: 30 μm, magnification: ×200).

**Figure 4 ijms-25-00316-f004:**
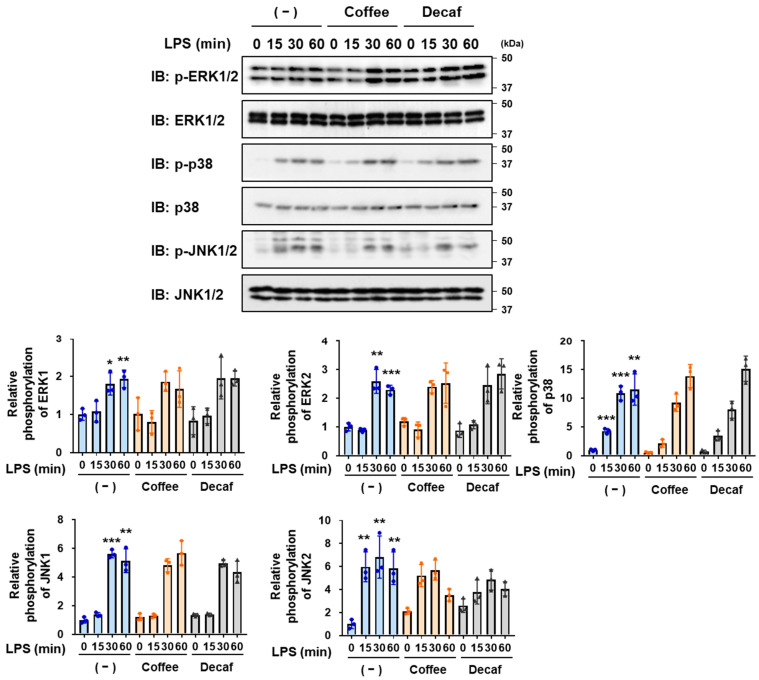
Coffee and decaffeinated coffee both have no effect on the LPS-induced phosphorylation of MAPK family in BV-2 cells. BV-2 cells were pretreated with coffee or decaffeinated coffee (Decaf) decoctions (5% (*v*/*v*)) for 1 h and were then stimulated with LPS (1 μg/mL) for the indicated periods. Immunoblotting was performed, and the relative phosphorylation level of ERK, p38, and JNK is shown in the graphs. Values are the mean ± S.D. of three independent experiments. * *p* < 0.05, ** *p* < 0.01, and *** *p* < 0.001 indicate significant differences from control cells.

**Figure 5 ijms-25-00316-f005:**
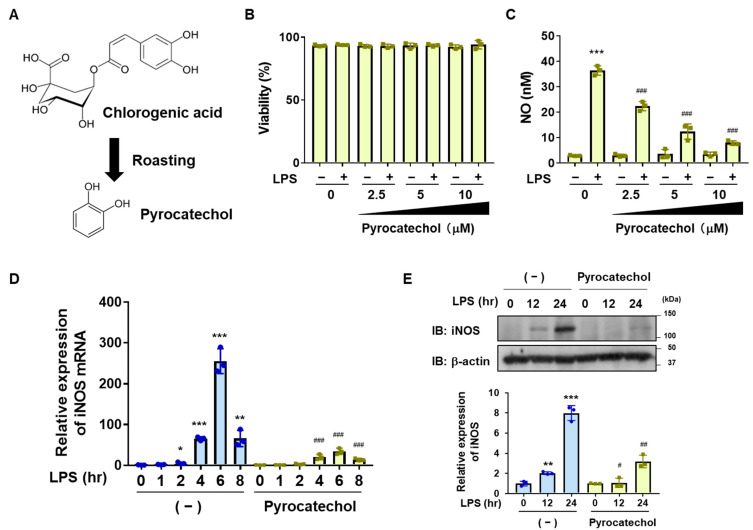
The coffee ingredient, pyrocatechol, inhibits LPS-induced NO production and expression of iNOS in BV-2 cells. (**A**) The scheme shows the reaction producing pyrocatechol by roasting coffee beans. (**B**,**C**) BV-2 cells were pretreated with pyrocatechol (2.5, 5, and 10 μM) for 1 h followed by stimulation with LPS (1 μg/mL) for 12 h. (**B**) Cell viability was measured. (**C**) The amount of NO in the culture supernatant was measured. Values are the mean ± S.D. of three independent experiments. *** *p* < 0.001 indicates significant differences from control cells stimulated with LPS. (**D**,**E**) BV-2 cells were pretreated with pyrocatechol (5 μM) for 1 h followed by stimulation with LPS (1 μg/mL) for the indicated periods. (**D**) The expression of iNOS mRNA was analyzed by RT-PCR. (**E**) The expression of iNOS was examined by immunoblotting, and the relative expression level of iNOS is shown in the graph. * *p* < 0.05, ** *p* < 0.01, and *** *p* < 0.001 indicate significant differences from control cells. ^#^
*p* < 0.05, ^##^
*p* < 0.01, and ^###^
*p* < 0.001 indicate significant differences from control cells treated with LPS.

**Figure 6 ijms-25-00316-f006:**
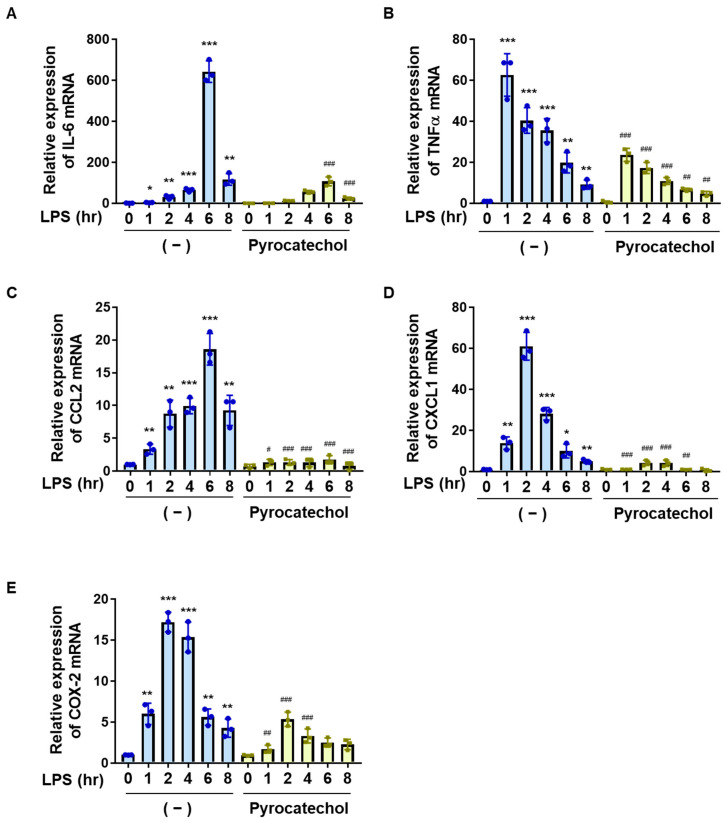
Pyrocatechol inhibits LPS-induced expression of inflammatory mediators in BV-2 cells. BV-2 cells were pretreated with pyrocatechol (2.5, 5, and 10 μM) for 1 h followed by stimulation with LPS (1 μg/mL) for 12 h. The mRNA expression of (**A**) IL-6, (**B**) TNFα, (**C**) CCL2, (**D**) CXCL, and (**E**) COX-2 was analyzed by RT-PCR. Values are the mean ± S.D. of three independent experiments. * *p* < 0.05, ** *p* < 0.01, and *** *p* < 0.001 indicate significant differences from control cells. ^#^
*p* < 0.05, ^##^
*p* < 0.01, and ^###^
*p* < 0.001 indicate significant differences from control cells treated with LPS.

**Figure 7 ijms-25-00316-f007:**
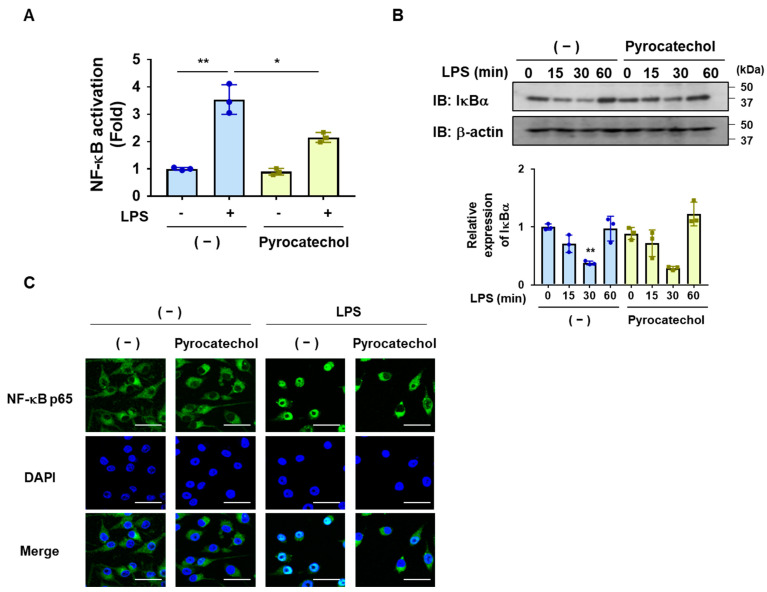
Pyrocatechol inhibits the LPS-induced nuclear localization and transcriptional activation of NF-κB in BV-2 cells. (**A**) BV-2 cells were transfected with pNF-κB-Luc and pRL-TK as described in the Methods. Cells were pretreated with pyrocatechol (5 μM) for 1 h followed by stimulation with LPS (1 μg/mL) for 8 h and Luciferase assay was performed. Values are the mean ± S.D. of three independent experiments. * and ** indicate *p* < 0.05 and *p* < 0.01, respectively. (**B**) BV-2 cells were pretreated with pyrocatechol (5 μM) for 1 h followed by stimulation with LPS (1 μg/mL) for indicated periods. The expression of IκBα was examined by immunoblotting and the relative expression level of IκBα is shown in the graphs. Values are the mean ± S.D. of three independent experiments. ** *p* < 0.001 indicates significant difference from control cells. (**C**) BV-2 cells were pretreated with pyrocatechol (5 μM) for 1 h followed by a stimulation with LPS (1 μg/mL) for 30 min. The localization of NF-κB p65 was visualized with an antibody (green). DAPI (blue) was also applied to visualize nuclei. (Scale bar: 30 μm, magnification: ×200).

**Figure 8 ijms-25-00316-f008:**
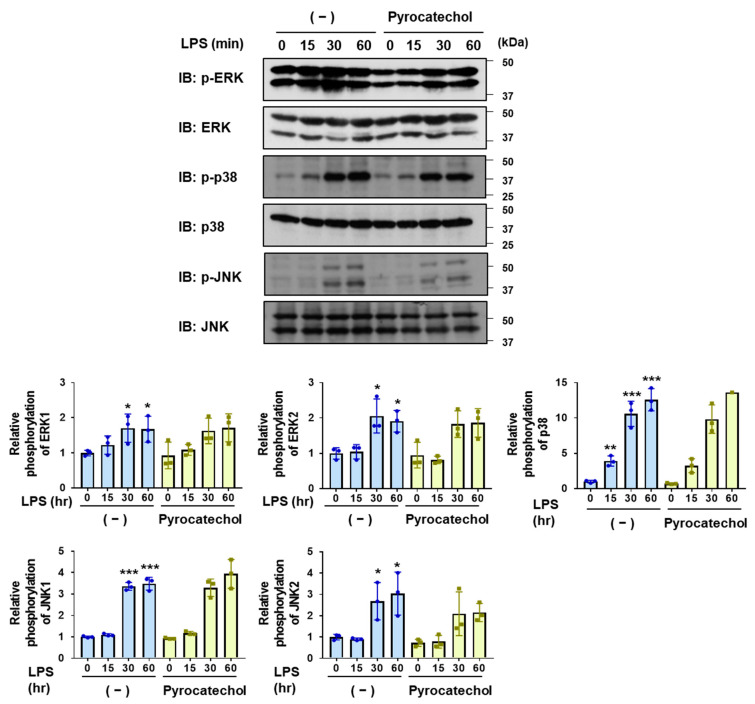
Pyrocatechol has no effect on the LPS-induced phosphorylation of the MAPK family in BV-2 cells. BV-2 cells were pretreated with pyrocatechol (5 μM) for 1 h and then stimulated with LPS (1 μg/mL) for the indicated periods. Immunoblotting was performed and the relative phosphorylation level of ERK, p38, and JNK is shown in the graphs. Values are the mean ± S.D. of three independent experiments. * *p* < 0.05, ** *p* < 0.01, and *** *p* < 0.001 indicate significant differences from control cells.

**Figure 9 ijms-25-00316-f009:**
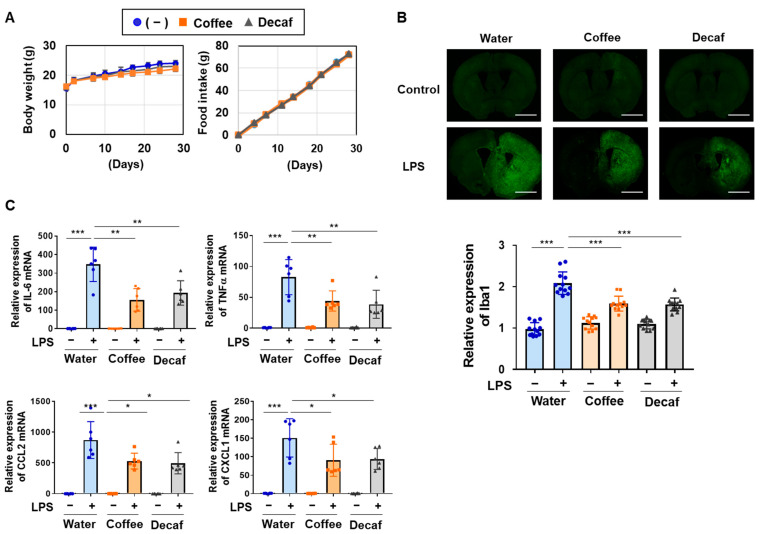
Coffee and decaffeinated coffee attenuate the LPS-induced accumulation of microglia and mRNA expression of IL-6, TNFα, CCL2, and CXCL1 in the murine brain. C57BL/6 mice (male, 4 weeks, *n* = 6) were given water, a 60% (*v*/*v*) coffee decoction, or 60% (*v*/*v*) decaffeinated coffee (Decaf) decoction for 4 weeks. (**A**) The body weight and food intake of each mouse were measured. (**B**) After 4 weeks of the intake of coffee or decaffeinated coffee (Decaf), LPS was injected into the right striatum of mice (*n* = 6). Seven days after the LPS injection, murine brain sections were prepared and two tissue sections obtained from six mice in each group were immunostained using anti-Iba1 antibody (Scale bar: 1 mm), and the luminance of each sample was measured by ImageJ. Data are presented as the mean ± SD of three independent experiments in the graph. *** indicates *p* < 0.001. (**C**) Six hours after the LPS injection, total RNA was extracted from murine brains. The mRNA expression of IL-6, TNFα, CCL2, and CXCL1 was analyzed by RT-PCR. Values are the mean ± S.D. of six independent experiments. *, **, and *** indicate *p* < 0.05, *p* < 0.01, and *p* < 0.001, respectively.

**Figure 10 ijms-25-00316-f010:**
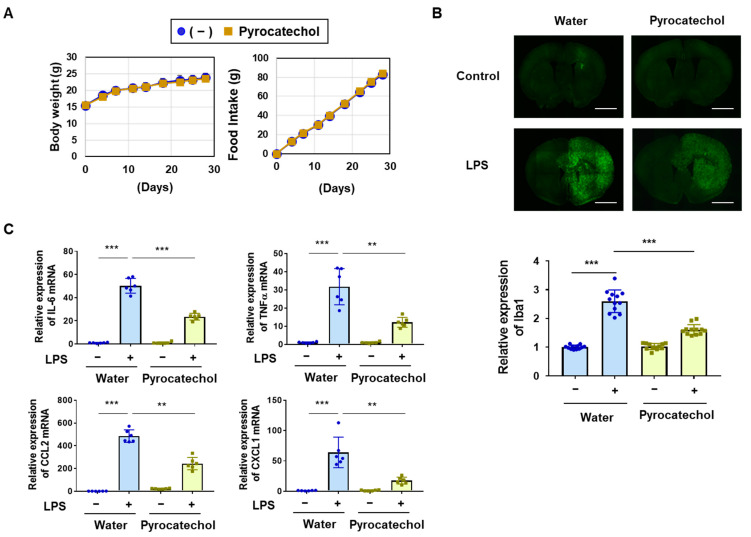
The intake of pyrocatechol attenuates the LPS-induced accumulation of microglia and mRNA expression of IL-6, TNFα, CCL2, and CXCL1 in the murine brain. C57BL/6 mice (male, 4 weeks, *n* = 6) were given water or 60 μM pyrocatechol for 4 weeks. (**A**) The body weight and food intake of each mouse were measured twice each week. (**B**) After 4 weeks of the intake of water or pyrocatechol, LPS was injected into the right striatum of the brain (*n* = 6). Seven days after the LPS injection, murine brain sections were prepared. Two tissue sections obtained from six mice in each group were immunostained using anti-Iba-1 atibody (Scale bar: 1 mm), and the luminance of each sample was measured by ImageJ. Data are presented as the mean ± SD of three independent experiments in the graph. *** indicates *p* < 0.05. (**C**) Six hours after the LPS injection, total RNA was extracted from murine brains. The mRNA expression of TNFα, IL-6, CCL2, and CXCL1 was analyzed by RT-PCR. Values are the mean ± S.D. of six independent experiments. ** and *** indicate *p* < 0.01 and *p* < 0.001, respectively.

**Figure 11 ijms-25-00316-f011:**
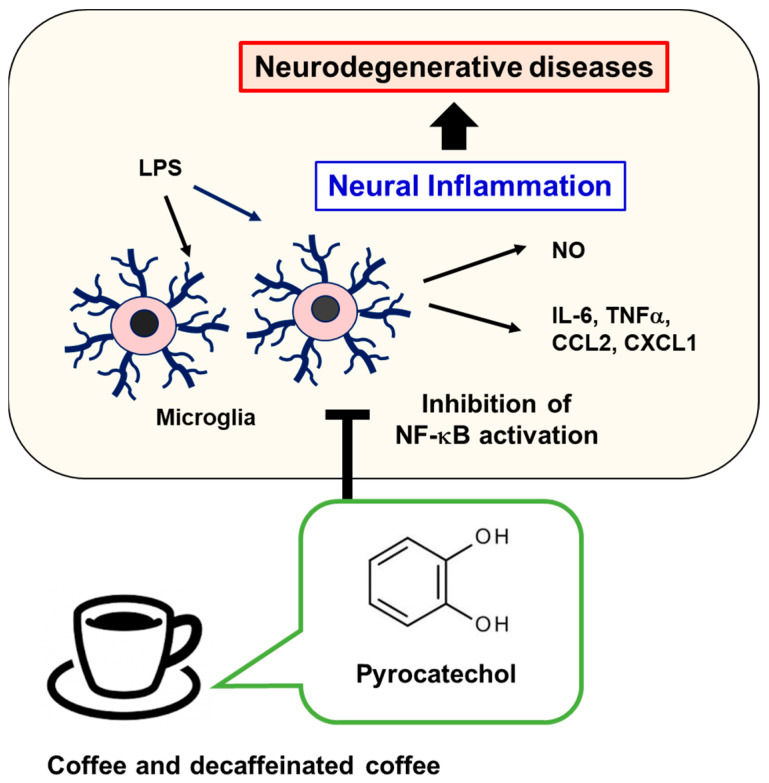
Pyrocatechol, a coffee ingredient, suppresses neuroinflammation. Pyrocatechol, a coffee ingredient, suppresses neuroinflammation by inhibiting the activation of NF-κB in microglia. Therefore, the intake of coffee is expected to suppress the onset of various neurodegenerative diseases.

## Data Availability

The data that support the findings of this study are available from the corresponding author upon reasonable request.

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
