# Peer review of "Suppression of Neuroinflammation by Coffee Component Pyrocatechol via Inhibition of NF-κB in Microglia"

_ijms, 2023, doi:10.3390/ijms25010316_

Round 1
Reviewer 1 Report
Comments and Suggestions for Authors
There is high demand for the development of innovative, inexpensive, and effective anticancer treatments using natural resources. Natural compounds have been increasingly discovered and used for cancer or neurodegeneration therapy owing to their high molecular diversity, novel biofunctionality, and minimal side effects. These compounds can be utilized as chemopreventive agents because they can efficiently inhibit cell growth, control cell cycle progression, and block several tumor-promoting signaling pathways. PI3K is an important upstream protein of the PI3K-Akt-mTOR pathway and a well-established cancer therapeutic target. Moreover, Keap1-Nrf2 is a fundamental signaling cascade known to promote or prevent carcinogenesis. Extensive studies identify the key target of modulatory aspects of Keap1-Nrf2 signaling against degenerative diseases. Nutraceuticals are those dietary agents with many health benefits that have immense potential for chemoprevention. The nutritional supplements known as nutraceuticals are found to be one of the most promising chemoprevention agents. Upon investigating the dual nature of Nrf2, it became clear that, in addition to shielding normal cells from numerous stresses, Nrf2 may also promote the growth of tumors. A wide variety of natural products have been widely used in chemoprevention therapy because they have antioxidant, anti-inflammatory, and anticancer activity. For instance, sulforaphane is the most extensively studied compound in modulating Keap1-Nrf signaling. Even though there is much evidence at preclinical levels, however further high-quality research is still required to validate the potential role of these nutraceuticals in Keap1-Nrf2 modulation-linked chemoprevention. Thus, interplay and coordination of redox interactions and their interaction with endogenous and exogenous antioxidant defence systems is an emerging area of reserach interest in anti-inflammatory anti-degenerative therapeutics. This reviewer is satisfied with the significance of this study, the care in which the study was performed, and the implications of the results for human health. Results presented are interesting and the questions posed are of extremely high interest, thus the paper does give adequate definitive information. Pending minor points, this paper can be accepted.
Minor concerns:
Given the relationship between vitagene network and its possible biological relevance in the defense mechanisms against oxidative stress-driven degenerative diseases or cancer, Authors can mention in the discussion appropriately this aspect (See and quote please doi: 10.1002/jnr.23925; doi: 10.3390/nu11102417).
english is ok
Author Response
We sincerely appreciate the constructive suggestion by reviewer. We additionally described following discussion with citing suggested references [49, 50] and additional reference [51]. (l. 421-430)
“In the previous study, we additionally clarified that the inhibitory effect of pyrocatechol on the activation of NF-kB was due to the pyrocatechol-induced activation of Keap1-Nrf2 pathway in macrophages [32]. It is quite simple speculation that similar machinery regulating NF-kB activation may work in also BV-2 cells. Several studies have reported that oxidative stresses-induced activation of Nrf2 most likely contributes to the prevention of not only neural inflammation but also tumorigenesis and tumor progression [49, 50]. Inflammation-related cells such as macrophages are well known to present in tumor microenvironment and contribute to tumor progression [51]. Accordingly, although it maybe too speculative, coffee decoction and its ingredients such as pyrocatechol may also suppress tumorigenesis or tumor progression.”
Reference
- Basak U et al. Tumor-associated macrophages: an effective player of the tumor microenvironment. Front Immunol. 2023 14: 1295257.
Reviewer 2 Report
Comments and Suggestions for Authors
Reviewer Comments:
The manuscript entitled “Suppression of neuroinflammation by coffee component pyrocatechol via inhibition of NF- kB in microglia” postulates that coffee consumption suppresses neuroinflammation, which is closely linked to the development of neurodegenerative diseases. While the authors have conducted commendable research, specific points merit consideration before advancing to the next stage. The following are points and suggestions for your consideration:
1. The manuscript could benefit from exploring the health impact of regular coffee consumption and its products. This addition would enhance the overall value of the study.
2. Discussing the correlation between sleep patterns and coffee consumption in the manuscript may be significant, as this relationship is noteworthy in the context of neurodegenerative diseases.
3. The manuscript lacks a thorough discussion of its results compared to recently published articles. Strengthening this aspect would support the manuscript.
4. Could the authors provide more insights into the downstream effects of NF-κB inhibition, particularly concerning changes in the expression of other genes involved in inflammation?
5. Was there an exploration of the potential off-target effects of pyrocatechol on cellular processes beyond inflammation? Including this information would enhance the accuracy of the study.
6. Were any adverse effects observed in the mice treated with pyrocatechol? Reporting on potential adverse effects is essential for a comprehensive understanding of the intervention.
7. Is there a concentration range at which pyrocatechol may lose its anti-inflammatory effects or exhibit toxicity? Elaborating on this aspect would contribute valuable information.
8. Are there any limitations or considerations concerning the consumption of pyrocatechol in the context of human health? Addressing this question would provide a more nuanced understanding of the potential implications of the study.
9. If the authors discuss any study limitations, such as the specific cell line used, the concentration of LPS, or the model system employed? Acknowledging and addressing limitations adds transparency to the research.
10. Considering the points raised, I recommend a minor revision to further address these considerations and strengthen the manuscript.
Author Response
- The manuscript could benefit from exploring the health impact of regular coffee consumption and its products. This addition would enhance the overall value of the study.
Answer:
We sincerely appreciate the comment by reviewer. Following the comment by reviewer, we additionally describe followed sentences in Discussion. (l.346-355)
“To date, many epidemiological studies have reported the effects of coffee drinking on human health. It has been reported that coffee consumption is well correlated with the prevention of onsets of various type of cancers including colorectal cancer and pancreatic cancer [38, 39]. It was also shown that coffee consumption enhances the efficiency of the treatment with tamoxifen to breast cancer patients [40]. We found that coffee decoction accelerated the ability of tamoxifen to induce the apoptotic cell death in breast cancer cell line, MCF-7 cells through the arrest of cell cycle and activation of p53 tumor suppressor [41]. In addition, we also reported that coffee decoction possesses the ability to suppress the insulin-caused adipocyte differentiation [36], suggesting that the coffee maybe also effective onto the prevention of obesity.
References
- Budhathoki et al. Coffee intake and the risk of colorectal adenoma: The colorectal adenoma study in Tokyo. Int J Cancer. 2015, 37,463-70.
- Nishi et al. Dose-response relationship between coffee and the risk of pancreas cancer. Jpn J Clin Oncol. 1996, 26, 42-48.
- Rosendahl et al. Caffeine and Caffeic Acid Inhibit Growth and Modify Estrogen Receptor and Insulin-like Growth Factor I Receptor Levels in Human Breast Cancer. Clin Cancer Res. 2015, 21, 1877-1887.
- Funakoshi-Tago et al. Coffee decoction enhances tamoxifen proapoptotic activity on MCF-7 cells. Sci Rep. 2020, 10, 19588.
- Discussing the correlation between sleep patterns and coffee consumption in the manuscript may be significant, as this relationship is noteworthy in the context of neurodegenerative diseases.
Answer:
As suggested by reviewer, it is well known that the consumption of caffeine affects the sleep pattern, therefore, it would be required to discuss about the relationship between coffee consumption, sleep pattern and prevalence of neurodegenerative diseases. However, it was reported that prevalence of REM sleeping behavior disorder is increased during the progression of Parkinson’s disease [46]. In this article, the authors concluded that REM sleeping behavior disorder seems to be a progression marker of Parkinson’s disease. According to this report, disorder of sleep pattern most unlikely affect the onset of neurodegenerative diseases.
We added sentences described above into discussion. (l. 371-377)
Reference
- Sixel-Döring et al. The Increasing Prevalence of REM Sleep Behavior Disorder with Parkinson's Disease Progression: A Polysomnography-Supported Study. Mov Disord Clin Pract. 2023, 10, 1769-1776.
- The manuscript lacks a thorough discussion of its results compared to recently published articles. Strengthening this aspect would support the manuscript.
Answer:
We really agree with the constructive suggestion by reviewer. We additionally referred two articles published in 2023. Then, we discussed about the results reported by these articles, and compared them to our current data. We additionally described followed sentences in discussion. (l.431-447)
“Recently, several studies also reported that ingredients or their derivatives derived from foods or plants suppress neural inflammation responses in microglia or its related neurodegenerative diseases. Using in vitro and in vivo experimental model, Bao and colleagues reported that Shikimic acid suppressed LPS-induced neuroinflammation through the inhibition of NF-kB activation [52]. They further described that shikimic acid activated AKT-Nrf2 pathway to inhibit NF-kB, however they failed to show detailed mechanism how AKT is activated by shikimic acid, and Nrf2 is activated by AKT. Li et al. also clarified that neferine exhibited anti-inflammatory activity in BV-2 cells through inhibition of LPS-induced phosphorylation and nuclear translocation of the NF-kB p65 subunit [53]. Comparing to these reports, we also clarified that pyrocatechol suppresses the nuclear translocation and the transcriptional activation of NF-kB in LPS-stimulated BV-2 cells. For the activation of Nrf2, inactivation of ubiquitination machinery including KEAP1 is known to be essential, and we also clarified that inactivation of KEAP1 is indispensable step for pyrocatechol-induced activation of Nrf2 [54]. In our study, it is still unclear how pyrocatechol suppresses the function of NLS of NF-kB p65 subunit, and how pyrocatechol causes the inactivation of KEAP1. These should be clarified to understand the biological activities in coffee decoctions in future investigation.”
References
- Bao et al. Shikimic acid (SA) inhibits neuro-inflammation and exerts neuroprotective effects in an LPS-induced in vitro and in vivo model. Front Pharmacol. 2023, 14, 1265571.
- Li et al. Neferine exerts anti‑inflammatory activity in BV‑2 microglial cells and protects mice with MPTP‑induced Parkinson's disease by inhibiting NF‑κB activation. Mol Med Rep. 2023, 28, 235.
- Kobayashi et al. Unique function of the Nrf2-Keap1 pathway in the inducible expression of antioxidant and detoxifying enzymes., Methods Enzymol. 2004, 378, 273-286.
- Could the authors provide more insights into the downstream effects of NF-κB inhibition, particularly concerning changes in the expression of other genes involved in inflammation?
Answer:
We observed the effect of coffee on the expression of COX-2 and added the data in Figure 2 and Figure 6. We explained COX-2 in introduction by quoting reference and added the sequence of COX-2 primer and he following sentences.
(l. 47-48, l. 382-388, l.503-504)
“We showed that coffee decoction suppressed the expression of LPS-induced expression of NF-kB-target genes such as IL-6, TNFa, CCL2, CXCL1 and COX-2 in BV-2 cells (Figure 2 and Figure 6). On the other hand, we observed that the expression of Interleukin-10 (IL-10), which has the function of suppressing inflammatory responses, was hardly changed by LPS stimulation or the treatment with coffee decoction or pyrocatechol [32], suggesting that the anti-inflammatory activity of coffee decoction is due to inhibition of NF-kB activity.”
References
- Salvemini et al. Endogenous nitric oxide enhances prostaglandin production in a model of renal inflammation. J Clin Invest. 1994, 93, 1940-1947.
- D'Acquisto et al. Involvement of NF-kappaB in the regulation of cyclooxygenase-2 protein expression in LPS-stimulated J774 macrophages. FEBS Lett. 1997, 418, 175-178.
- Was there an exploration of the potential off-target effects of pyrocatechol on cellular processes beyond inflammation? Including this information would enhance the accuracy of the study.
Answer:
Thank you for your valuable opinion. I have added the following sentences to the discussion as an issue that should be considered in the future.
(l. 414-420)
“Furthermore, we have not yet investigated the potential off-target effects of pyrocatechol on cellular processes beyond inflammation. In this study, we clarified that (1) coffee and pyrocatechol did not affect the cell growth and viability of BV-2 cells as shown in Figure 1 and Figure 5 and that (2) coffee and pyrocatechol suppressed the nuclear translocation and transcriptional activation of NF-kB. We clarified that pyrocatechol targets at least two machineries, however we would find something else in future investigation.”
- Were any adverse effects observed in the mice treated with pyrocatechol? Reporting on potential adverse effects is essential for a comprehensive understanding of the intervention.
Answer:
In our experimental system, no obvious side effects were observed when administering pyrocatechol to mice. As suggested by the reviewer, we have written about the toxicity of pyrocatechol that has been reported, and compared the doses in this experiment.
We wrote about it in detail in the answer to comment 8. (l. 409-414)
- Is there a concentration range at which pyrocatechol may lose its anti-inflammatory effects or exhibit toxicity? Elaborating on this aspect would contribute valuable information.
Answer:
We added the following sentences in Results. (l. 210-212)
“ The results of examining the concentration of pyrocatechol showed that above 0.6 μM, pyrocatechol significantly inhibited LPS-induced NO production, and that pyrocatechol up to 100 μM did not exhibit cytotoxicity after 12 h of treatment.”
- Are there any limitations or considerations concerning the consumption of pyrocatechol in the context of human health? Addressing this question would provide a more nuanced understanding of the potential implications of the study.
Answer:
Thank you so much for constructive comment. We added the sentences described below in discussion. (l. 409-414)
“Previous studies reported that pyrocatechol exhibited acute toxicity and mutagenicity which could cause oncogenicity [48]. Although we understand that the concentration of pyrocatechol utilized in our study did not exhibit the toxicity to the mice, we still need to confirm its mutagenicity. To overcome this problem, we need to chemically modify pyrocatechol to develop safe derivatives.”
Reference
- Final report of the safety assessment of hydroquinone and pyrocatechol, J. Amer. Coll. Toxicol, 1986, 5, 123-165
- If the authors discuss any study limitations, such as the specific cell line used, the concentration of LPS, or the model system employed? Acknowledging and addressing limitations adds transparency to the research.
Answer:
We discussed about study limitations in the section of discussion as following in revised manuscript. (l. 448-454)
“In the present study, we investigated the LPS-treated microglial cell line BV-2 and mice injected with LPS into the brain to show that coffee and its component pyrocatechol suppressed neuroinflammation (Figure 11). However, it is considered that actual neural inflammations related to neurodegenerative diseases including AD and PD are triggered by the accumulation of aggregated Ab or a-synuclein [42, 43]. In our current experimental system, it is difficult to test whether pyrocatechol is also effective on neural inflammation caused by Ab or a-synuclein, and this issue needs to be resolved in the future.”
- Considering the points raised, I recommend a minor revision to further address these considerations and strengthen the manuscript
Answer:
We have addressed the above points as much as possible. We feel that our manuscript has been greatly improved thanks to the valuable comments from reviewers.